# Experimental and computational investigation of enzyme functional annotations uncovers misannotation in the EC 1.1.3.15 enzyme class

**Elzbieta Rembeza**, **Martin K. M. Engqvist***

Department of Biology and Biological Engineering, Chalmers University of Technology, Gothenburg, Sweden

* martin.engqvist@chalmers.se

**Data Availability Statement:** Experimental data as well as database files are available for download from Zenodo (DOI: 10.5281/zenodo.4518801, https://zenodo.org/record/4518801).

## Abstract

Only a small fraction of genes deposited to databases have been experimentally characterised. The majority of proteins have their function assigned automatically, which can result in erroneous annotations. The reliability of current annotations in public databases is largely unknown; experimental attempts to validate the accuracy within individual enzyme classes are lacking. In this study we performed an overview of functional annotations to the BRENDA enzyme database. We first applied a high-throughput experimental platform to verify functional annotations to an enzyme class of S-2-hydroxyacid oxidases (EC 1.1.3.15). We chose 122 representative sequences of the class and screened them for their predicted function. Based on the experimental results, predicted domain architecture and similarity to previously characterised S-2-hydroxyacid oxidases, we inferred that at least 78% of sequences in the enzyme class are misannotated. We experimentally confirmed four alternative activities among the misannotated sequences and showed that misannotation in the enzyme class increased over time. Finally, we performed a computational analysis of annotations to all enzyme classes in the BRENDA database, and showed that nearly 18% of all sequences are annotated to an enzyme class while sharing no similarity or domain architecture to experimentally characterised representatives. We showed that even well-studied enzyme classes of industrial relevance are affected by the problem of functional misannotation.

## Author summary

Correct annotation of genomes is crucial for our understanding and utilization of functional gene diversity, yet the reliability of current protein annotations in public databases is largely unknown. In our work we validated annotations to an S-2-hydroxyacid oxidase enzyme class (EC 1.1.3.15) by assessing activity of 122 representative sequences in a high-throughput screening experiment. From this dataset we inferred that at least 78% of the sequences in the enzyme class are misannotated, and confirmed four alternative activities among the misannotated sequences. We showed that the misannotation is widespread throughout enzyme classes, affecting even well-studied classes of industrial relevance.

**Funding:** The author(s) received no specific funding for this work.

**Competing interests:** The authors have declared that no competing interests exist.

Overall, our study highlights the value of experimental and computational validation of predicted functions within individual enzyme classes.

## Introduction

With the steady increase of genetic information deposited to public databases, the proportion of experimentally characterised sequences continues to decline. At the time of writing the UniProt/TrEMBL protein database contains nearly 185 million entries, with only 0.3% of them having been manually annotated and reviewed in the Swiss-Prot database [1]. Furthermore, the experimentally characterised sequence diversity is limited, representing proteins mainly from eukaryotes and model organisms. As the traditional experimental methods for determining protein function cannot keep up with the increase in genomic data, high-throughput methods enabling protein family-wide substrate profiling for hundreds of enzymes are being implemented. Data generated in such approaches are important for understanding sequence-function relationships in the tested protein families; they have led to the discovery of novel enzymatic activities as well as identified enzymes with diverse physicochemical properties [2–6]. Additionally, several global initiatives have been undertaken to bring together computational and experimental scientists to accelerate discovery of novel protein activities and enable more trustworthy functional annotations [7–9].

In spite of new platforms enabling more efficient experimental protein characterisation, automated annotation methods form the basis for functional assignment of new proteins [10]. These methods commonly rely on inferring a function from sequence similarity to curated sequences or to already existing entries in a given database. Annotations can be transferred either as a free text description of a function, or as more structured vocabularies like Gene Ontology [11] or Enzyme Commision classifications. Sequence similarity-based annotation pipelines enable processing of vast amounts of newly sequenced data, however, it has been shown that if not applied appropriately, they result in erroneous functional annotations, which later percolate throughout databases [12–18]. In order to improve functional annotations and predict novel protein subtypes, more refined methods are constantly being developed. They exploit signatures of protein families and domains, orthology, patterns of functional divergence, or a mixture of all the approaches [19–22]. Still, the quality of functional annotations is considered far from perfect [23]. Existing reports on the misannotation issue in public databases estimate the annotation error to be between 5–80%, depending on the protein family and database, and indicate overprediction as the main cause of such errors [24,25]. It is worth noting that these reports are based on entries from over 15 years ago, before a rapid increase in genome sequencing projects caused by the rise of low-cost sequencing technologies. The reliability of present annotations in public databases is largely unknown.

In this study we utilize a high-throughput experimental platform, similar to those used for substrate profiling of protein families, to verify functional annotations to an enzyme class in the BRENDA database [26]. We provide an overview of all the sequences annotated as S-2-hydroxyacid oxidases (EC 1.1.3.15) and select 122 representatives of the class for experimental screening of their predicted function. We show that the majority of the sequences contain non-canonical protein domains, do not catalyse the predicted reaction, and are wrongly annotated to the enzyme class. Among the misannotated sequences we confirm four alternative enzymatic activities. Finally, a computational analysis of all EC classes in BRENDA reveals that a large proportion of sequences are annotated to enzyme classes with no similarity to characterised enzymes, a problem which warrants further investigation.

## Results

### Exploration of EC 1.1.3.15 sequence space

Enzyme Commission (EC) classification is a numerical classification system for enzymes, based on the chemical reaction they catalyse and substrate they act upon. Different enzymes catalysing the same reaction receive the same EC number, regardless of their similarity in sequence or structure.

A medium-size, easy to assay enzyme class 1.1.3.15 (S-2-hydroxyacid oxidase) was chosen for this proof-of-concept study. Representatives of the class oxidize the hydroxyl group of S-2-hydroxyacids like glycolate or lactate to 2-oxoacids, using oxygen as an electron acceptor (S1 Fig). All characterised enzymes of this class belong to a family of FMN-dependent α-hydroxy acid oxidases/dehydrogenases. Members of this protein family share high structural and functional similarities but differ in the ultimate electron acceptor: oxygen (S-2-hydroxyacid oxidase, EC 1.1.3.15; lactate monooxygenase, EC 1.13.12.4), cytochrome c (flavocytochrome b2, EC 1.1.2.3) or quinone (S-mandelate dehydrogenase, EC 1.1.99.31) [27–29]. A characteristic feature for S-2-hydroxyacid oxidases is their broad substrate scope *in vitro*, although the physiological substrate for plant and mammalian homologues is mainly glycolate or long chain hydroxyacids [30–32], while lactate is the main physiological substrate of bacterial homologues [33,34].

Members of EC 1.1.3.15 are of high biological importance, with plant GOX being crucial for photorespiration, mammalian HAOs taking part in glycine synthesis and fatty acid oxidation, and bacterial LOX metabolising L-lactate as an energy source [31]. Human HAO1 was recently proposed as a target for treating primary hyperoxaluria, an autosomal metabolic disorder leading to decline in renal function [35]. Bacterial LOX are of particular medical and industrial interest, being used for lactate biosensor development in clinical care, sport medicine, and food processing [36].

To obtain an overview of sequence diversity in EC 1.1.3.15, we downloaded all sequences annotated to this EC in BRENDA 2017.1 and obtained 1058 unique sequences after filtering out partial genes. The sequence interrelatedness of these diverse proteins was visualized in a multidimensional scaling (MDS) plot using computed UniRep embeddings [37]; a smaller distance in this plot indicates higher relatedness (Figs 1 and S2). Among the 1058 sequences 17 are characterised and/or manually curated enzymes: sequences listed in BRENDA [38] as experimentally tested or in SwissProt [1] as manually curated sequences having experimental evidence at protein level. Over 90% of the enzymes annotated to this enzyme class are of bacterial origin, nearly 6% of eukaryotic and 2.6% of archaeal (Fig 1A). Strikingly, 14 out of 17 characterised enzymes are of eukaryotic origin, showing a clear over-representation. The characterised sequences also cluster close together in the visualization (Fig 1A, 1B and 1C), indicating that the characterised/curated sequence diversity in EC 1.1.3.15 is limited.

We next determined the similarity of each sequence in EC 1.1.3.15 to the closest characterised S-2-hydroxyacid oxidase in terms of alignment-based sequence identity and domain architecture. Most sequences have little similarity with the characterised ones; 79% of sequences annotated as 1.1.3.15 share less than 25% sequence identity with the closest characterised/curated sequence (Figs 1B and S3). Furthermore, only 22.5% of the 1058 sequences are predicted to contain the FMN-dependent dehydrogenase domain (FMN_dh, PF01070) which is canonical for known 2-hydroxy acid oxidases (Fig 1C). The majority of sequences were predicted to contain non-canonical domains, such as FAD binding domains characteristic for FAD-dependant oxidoreductases (PF01266, PF01565, PF02913), as well as a cysteine rich domain (PF02754) and 2Fe-2S binding domain (PF04324). Many of the sequences with non-canonical domains form distinct clusters (Fig 1C). An analysis of alignment-based similarity

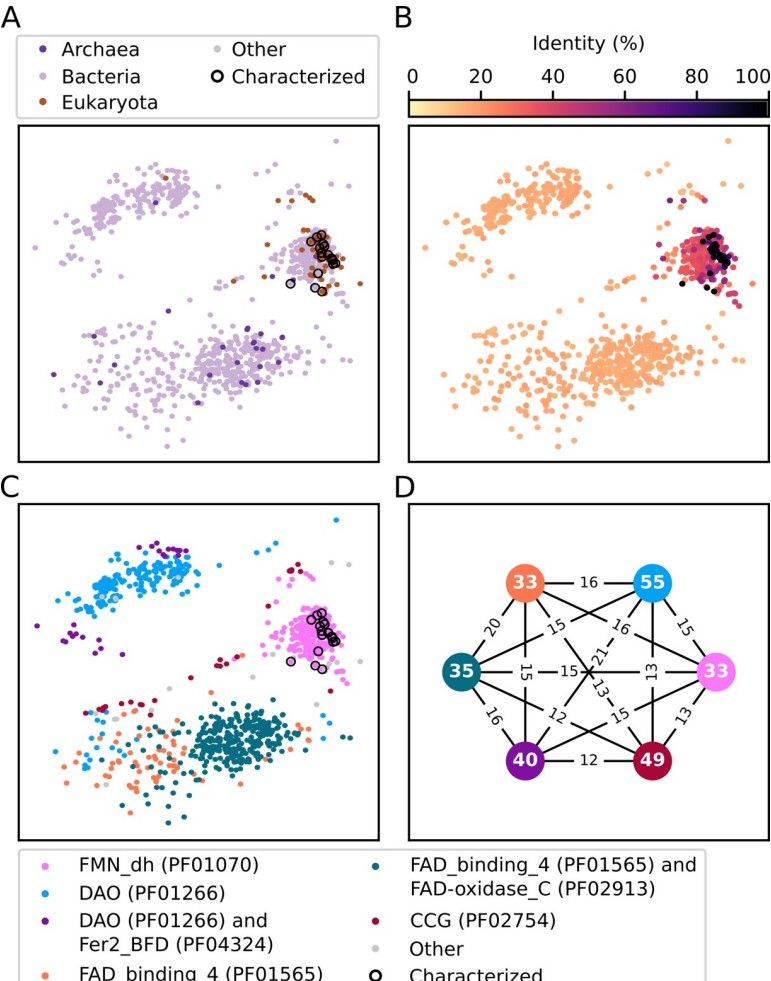

**Fig 1. Sequence space of sequences annotated to EC 1.1.3.15.** Sequences listed in BRENDA and SwissProt as experimentally tested are encircled **(A)** Taxonomic origin of sequences. **(B)** Percentage of sequence identity to the closest experimentally tested or curated S-2-hydroxyacid oxidase. **(C)** Pfam domain architecture. **(D)** The mean alignment-based sequence identity between and within domain clusters. Pfam protein domains: FMN_dh (PF01070)—FMN-dependent dehydrogenase, DAO (PF01266)—FAD dependent oxidoreductase, Fer2_BFD (PF04324)—BFD-like [2Fe-2S] binding domain, FAD_binding_4 (PF01565)—FAD binding domain, FAD-oxidase_C (PF02913)—FAD linked oxidases C-terminal domain, CCG (PF02754)—cysteine-rich domain.

between these domain clusters showed that the average sequence identity to the canonical FMN-dependent dehydrogenase domain cluster is below 16% for all clusters. An all versus all comparison revealed that no two clusters share more than 21% average sequence identity, while the identity of sequences within clusters ranges between 33% and 55% (Fig 1D).

This analysis clearly shows that the enzyme class EC 1.1.3.15 contains a set of very diverse protein sequences, the majority of which have low identity to sequences with experimental evidence, and also lack protein domains characteristic of S-2-hydroxy acid oxidases.

## Experimental characterisation of EC 1.1.3.15

Due to the large diversity of sequences annotated to EC 1.1.3.15 we carried on to experimental validation of their predicted activity. A total of 122 genes throughout the sequence space of the enzyme class were selected (S4 Fig, left panel), synthesised, cloned and recombinantly

expressed in *Escherichia coli* in a high throughput set up. Out of the 122 proteins, 65 were in soluble state (53%), with archaeal and eukaryotic proteins being proportionally less soluble than bacterial proteins (S4 Fig, right panel). Despite representing only half of the sequences chosen for experimental characterisation, the soluble proteins were still distributed throughout the sequence space of EC 1.1.3.15 (S4 Fig, left panel). The 65 soluble proteins were tested for S-2-hydroxy acid oxidase activity in an Amplex Red peroxide detection assay with a set of six 2-hydroxy acids: glycolate, lactate, 2-hydroxyoctanoate, 2-hydroxydecanoate, mandelate, and 2-hydroxyglutarate (S5 Fig).

## Characterisation of proteins carrying the canonical FMN-dh domain

We first investigated 24 proteins representing a cluster of 230 sequences containing the FMN_dh domain; these have the highest sequence identity to previously characterised 2-hydroxy acid oxidases (Figs 1C and 2A). Among them 14 proteins were active with a broad substrate range, as is characteristic for enzymes in EC 1.1.3.15, while 10 proteins were inactive. Bacterial sequences in the cluster were predominantly active with lactate, medium chain and aromatic 2-hydroxy acids, whereas the two active eukaryotic enzymes showed the highest activity with glycolate and lactate.

We next analysed whether the 24 investigated proteins contain the seven conserved amino acid residues involved in catalysis and substrate binding [32], both using a multiple sequence alignment and protein structure analysis (Fig 2A and 2B). In 12 of the 14 active proteins all seven residues are conserved (Fig 2A), whereas 8 of the 10 inactive proteins lack at least one of the conserved residues. Presence of the seven conserved amino acids is thus a strong–but not absolute–indication of S-2-hydroxyacid oxidase activity. As no investigation of folding of the purified proteins was performed, it is possible that the enzymes showing no activity, particularly the ones with all conserved residues present, were incorrectly folded.

The seven active site residues are, however, conserved not only in S-2-hydroxyacid oxidases, but also among all the members of the FMN-dependant S-2-hydroxyacid oxidase/dehydrogenase family [28]. We therefore looked for sequence motifs indicating the presence of other family members in our selection (Fig 2C). Two of the screened proteins (B8MKR3 and B8MMC0 from *Talaromyces stipitatus*) contain a heme binding domain (PF00173) characteristic for flavocytochrome b2 L-lactate dehydrogenase (EC 1.1.2.3) [29] (Figs 2A and S6). These two proteins were tested *in vitro* for their ability to reduce cytochrome c, a physiological electron acceptor of flavocytochrome b2 L-lactate dehydrogenase. Indeed, the B8MKR3 protein displayed cytochrome b2 L-lactate dehydrogenase activity (S7 Fig). Additionally, four other proteins (E6SCX5 from *Intrasporangium calvum*, C9Y9E7 from a *Curvibacter* species and W6W585 from *Rhizobium sp. CF080*) contain a longer stretch in loop 4 characteristic for S-mandelate dehydrogenase (EC 1.1.99.31) and L-lactate 2-monooxygenase (EC 1.13.12.4) [27,28] (Figs 2A and S6). As seen in our Amplex Red assay, the four proteins display a high activity with mandelate, suggesting their native function may be as S-mandelate dehydrogenases, although further experiments are needed to determine this.

Out of the 230 members of the FMN_dh cluster–with high sequence identity to previously characterised EC 1.1.3.15 enzymes–a total of 6 proteins (2.6%) are predicted to contain a heme binding domain and 50 (22%) contain a longer stretch in loop4, indicating that those sequences might be misannotated and would be better placed in other EC classes. However, a thorough biochemical and genetic characterisation of such enzymes is needed to test this hypothesis.

## Characterisation of proteins carrying non-canonical domains

Next, we investigated the activity of 41 proteins not containing the canonical FMN-dh domain (Fig 1C), yet representing a full 78% of all sequences annotated to EC 1.1.3.15 in BRENDA.

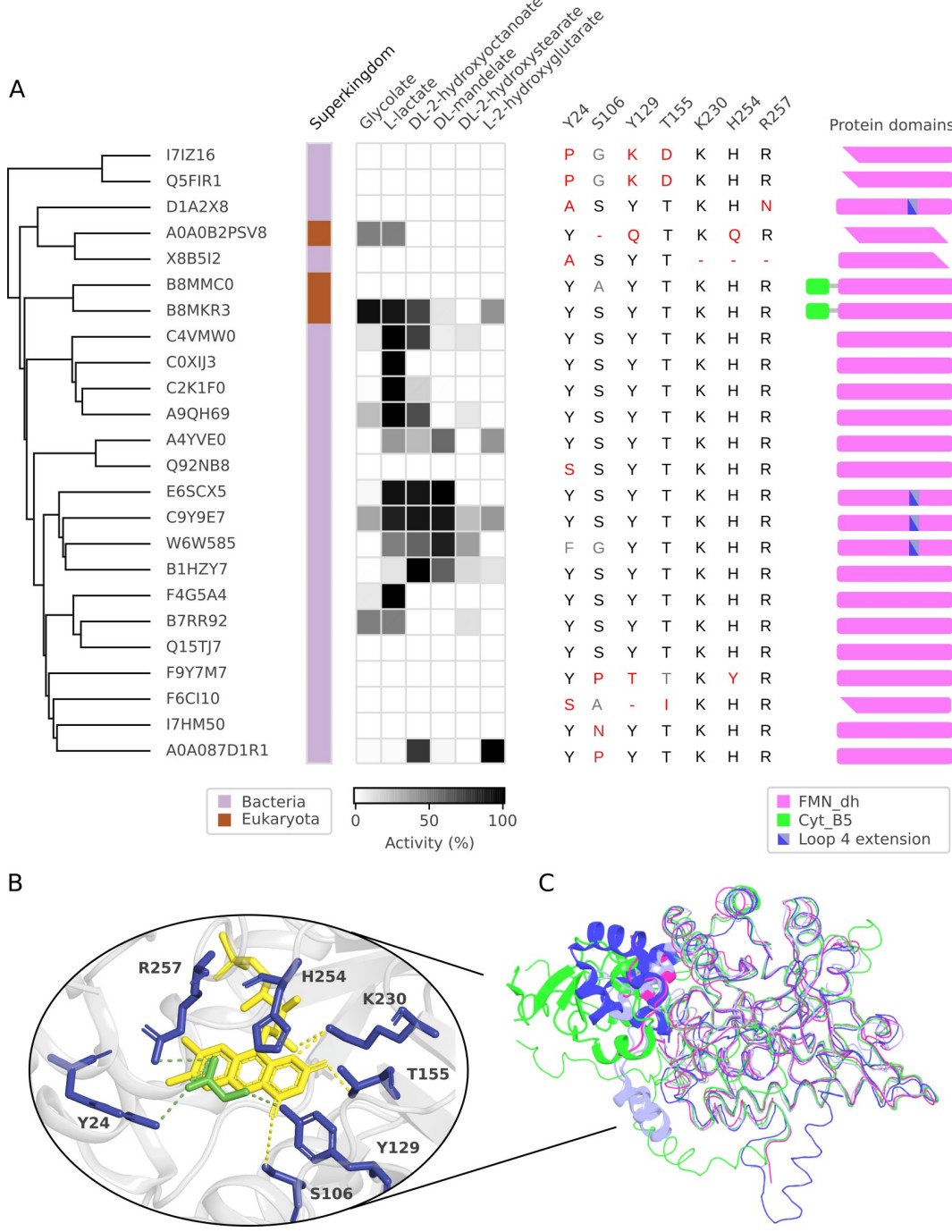

**Fig 2. Characterisation of protein cluster with high sequence identity to previously characterised S-2-hydroxyacid oxidases.** (A) Activity screen and protein characteristics. Dendrogram indicates protein relatedness. Superkingdoms: light purple—Bacteria, brown—Eukaryotes. Recorded activities are marked with squares, for proteins active with more than one substrate, the substrate preference is shaded with the highest activity for each enzyme scaled to 100%. Listed amino acids correspond to conserved residues in a glycolate oxidase from *S. oleracea*. The cartoons represent predicted domain and motif composition of the sequences, based on Pfam search. Domains lacking full Pfam alignment are represented with a sharp edge. FMN-binding domain (FMN_dh, PF01070) is marked in magenta, cytochrome b5-like heme binding domain (Cyt_B5, PF00173) is marked in green, and a prolonged stretch in loop4 is marked in blue. (B) Conserved amino acids of the active site of S-2-hydroxyacid oxidase mapped on a structure of glycolate oxidase from *S. oleracea* (PDB: 1GOX). Conserved residues are marked in blue, the FMN cofactor is marked in yellow, and the glycolate substrate in green. (C) Superimposed structures of the representatives of FMN-dependant 2-hydroxyacid oxidase/dehydrogenase family with their distinct motifs represented in a cartoon form: glycolate oxidase (magenta, PDB 1GOX), flavocytochrome b2 (green, PDB 1FCB), mandelate dehydrogenase (light blue, PDB 6BFG), lactate 2-monooxygenase (dark blue, PDB 6DVH).

These proteins have only low sequence identity with previously characterised S-2-hydroxyacid oxidases (Fig 1B and 1D).

Out of the 41 proteins, twelve come from the cluster predicted to contain a single FAD dependent oxidoreductase domain (DAO, PF01266). Six of the twelve solely oxidised the substrate L-2-hydroxyglutarate in the *in vitro* assay (Fig 3A). This narrow substrate scope is atypical for the previously known broad substrate-range EC 1.1.3.15 enzymes, which indicates an alternative native function of these proteins. Our findings are supported by those of a recent publication where activity of an *E. coli* homologue of the 6 DAO-containing proteins was described as L-2-hydroxyglutarate dehydrogenase (EC 1.1.99.2/ EC 1.1.5.13) [39]. As the Amplex Red activity assay used in our activity screen is designed to capture oxidase activity via hydrogen peroxide detection, we may have detected a low level of non-physiological oxidase

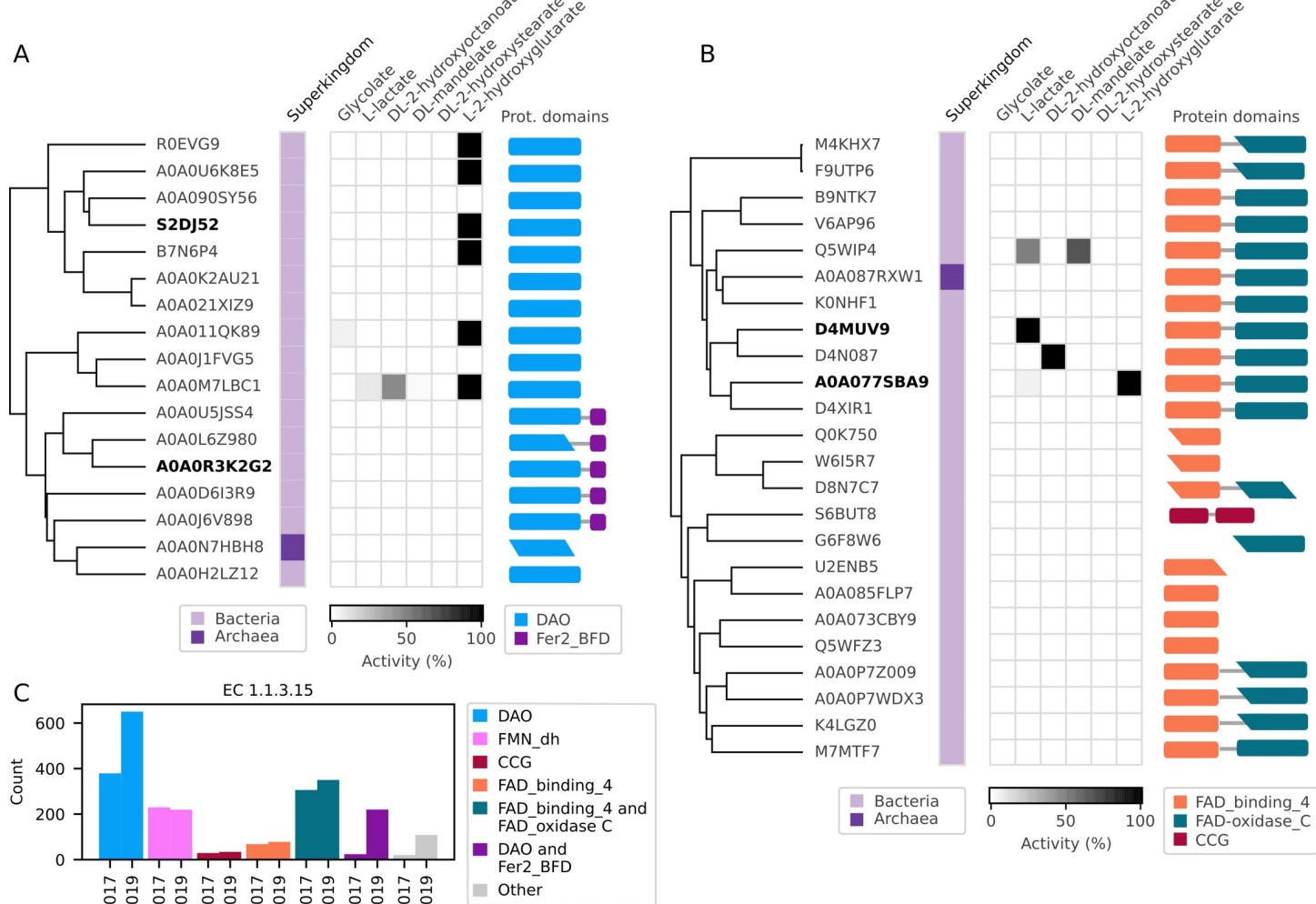

**Fig 3. Characterisation of protein clusters with low sequence identity to previously characterised S-2-hydroxyacid oxidases.** Dendrogram indicating protein relatedness. Superkingdoms: light purple—Bacteria, dark purple—Archaea. Activities are marked with squares; for proteins active with more than one substrate, the substrate preference is shaded. The cartoons represent predicted domain and motif composition of the sequences, based on Pfam search. Domains lacking full Pfam alignment are represented with a sharp edge. Proteins with alternative activities chosen for kinetic characterisation are marked in bold. **(A)** Characterisation of protein clusters containing DAO domain. FAD dependent oxidoreductase domain (DAO, PF01266) is marked in blue, BFD-like [2Fe-2S] binding domain (Fer2_BFD, PF04324) is marked in purple. **(B)** Characterisation of remaining protein clusters. FAD binding domain (FAD_binding_4, PF01565) is marked in orange, FAD linked oxidases C-terminal domain (FAD-oxidase_C, PF02913) is marked in green, cysteine rich domain (CCG, PF02754) is marked in red. **(C)** Comparison of Pfam domains of sequences annotated to EC 1.1.3.15 in BRENDA version 2017.1 and 2019.2.

activity of the 6 L-2-hydroxyglutarate dehydrogenases (see further discussion on the AR assay a few paragraphs below).

The remaining 29 sequences of the "non-canonical" clusters–containing either a BFD-like [2Fe-2S] binding domain (Fig 3A), or a FAD linked oxidases C-terminal domain, either alone or combined with a cysteine-rich domain (Fig 3B)–were either inactive or did not display consistent substrate preferences (Fig 3A and 3B). We hypothesised that due to the non-canonical domain architecture and low sequence identity to characterised enzymes, these proteins may catalyse reactions different from the ones initially tested. By searching database information regarding the Pfam [40] domains and combining this information with orthology-based annotations and literature search, we found that some of these sequences are similar to dehydrogenases operating on four distinct substrates: glycerol-3-phosphate, glycolate, D-lactate and D-2-hydroxyglutarate dehydrogenase.

In order to test whether the remaining 29 proteins catalyse these alternate reactions, we expressed and purified them, and the 22 successfully purified proteins were screened for the expected dehydrogenase activities with a set of common electron acceptors: nicotinamide adenine dinucleotide (NAD), nicotinamide adenine dinucleotide phosphate (NADP), the redox dye 2,6-Dichlorophenolindophenol (DCPIP), as well as the hydrogen peroxide probe Amplex Red (AR), and in selected cases cytochrome c (S8 Fig). When screened with DCPIP and AR, one protein was found to be active with glycerol-3-phosphate as a substrate (A0A0R3K2G2 from *Caloramator mitchellensis*), one with D-lactate (D4MUV9 from *Anaerostipes hadrus*) and one with D-2-hydroxyglutarate (A0A077SBA9 from *Xanthomonas campestris*). Additionally, three proteins (A0A0U5JSS4 from a *Clostridium* species, D4XIR1 from *Achromobacter piechaudii*, Q5WIP4 from *Bacillus clausii*) were active with each of the three substrates only in the AR screen (S8 Fig). None of the proteins were active with the electron acceptors NAD, NADP, or cytochrome c.

The fact that some of the tested enzymes show activity with both AR and DCPIP is counterintuitive as AR is a $H_2O_2$-dependent reporter, indicating that molecular oxygen is the electron acceptor, whereas DCPIP accepts electrons directly. Comparing standard curves of the two reporter molecules DCPIP and resorufin (the AR reaction product) revealed that the AR assay is several orders of magnitude more sensitive than DCPIP, on a molar basis (S9A Fig). We then carried out a direct comparison of enzyme activity in four purified enzymes using the DCPIP and AR assays. While the AR-dependent assay clearly gave the strongest signal, the enzymes displayed fifty to one hundred times higher catalytic rates in the DCPIP-based one (S9B Fig). Dehydrogenase activity is thus the prevalent one for the tested enzymes, although we were able to capture their trace oxidase activity.

Overall, our screen of the non-canonical clusters revealed their erroneous annotation as EC 1.1.3.15, and we found four alternative activities among those sequences: L-2-hydroxyglutarate dehydrogenase, D-2-hydroxyglutarate dehydrogenase, D-lactate dehydrogenase, and glycerol-3-phosphate dehydrogenase. Four representatives with the alternative activities were chosen for further characterization (Fig 3A and 3B, in bold); they were expressed, purified (S10A Fig), assayed at 25˚C and their kinetic parameters calculated (Table 1 and S10B Fig). Three of the

**Table 1. Kinetic parameters of selected proteins with functions distinct from S-2-hydroxyacid oxidase.** Values represent mean averages (+/- standard error of mean; n = 3).

| Enzyme | Substrate | $K_M$ [M] | $k_{cat}$ [s$^{-1}$] | $k_{cat}/K_M$ [M$^{-1}$s$^{-1}$] |
|---|---|---|---|---|
| **D4MUV9** | D-lactate | $0.40 +/- 0.04 \times 10^{-3}$ | 5.180 | $1.31 \times 10^4$ |
| **A0A077SBA9** | D-2-hydroxyglutarate | $0.08 +/- 0.01 \times 10^{-3}$ | 5.957 | $7.29 \times 10^4$ |
| **A0A077SBA9** | D-malate | $5.03 +/- 1.38 \times 10^{-3}$ | 0.039 | 7.78 |
| **S2DJ52** | L-2-hydroxyglutarate | $0.22 +/- 0.02 \times 10^{-3}$ | 3.719 | $1.68 \times 10^4$ |
| **A0A0R3K2G2** | glycerol-3-phosphate | $1.97 +/- 0.23 \times 10^{-3}$ | 0.242 | $1.23 \times 10^2$ |

four enzymes (D4MUV9, A0A077SBA9, S2DJ52) showed good catalytic efficiency with substrate affinities in the low micromolar range and $k_{cat}/K_M$ values above $1 \times 10^4 M^{-1}s^{-1}$, strengthening the possibility that these may be the natural substrates. Additionally, based on reports of a homologous protein [41], the protein A0A077SBA9 was screened and showed modest side activity with D-malate. The fourth enzyme, A0A0R3K2G2, showed affinity for glycerol-3-phosphate in the low millimolar range, but with $k_{cat}/K_M$ values approximately 100-fold lower than the other enzymes. Since this protein comes from the thermophilic bacterium *Caloramator mitchellensis*, whose optimal growth temperature is 55˚C, we speculate that its catalytic efficiency would be higher at higher experimental temperatures.

Taken together, our results indicate that proteins which do not contain the canonical FMN-dh domain, which represent 78% of all proteins annotated to EC 1.1.3.15 in BRENDA, likely have *in vitro* catalytic activities that do not match their current EC classification.

It is difficult to assess with certainty why these sequences were annotated to the EC 1.1.3.15 in the first place, and we can only speculate the origins of the misannotations. L-2-hydroxyglutarate dehydrogenase upon its discovery was incorrectly characterised as an oxidase [42], and thus received an incorrect assignment to EC 1.1.3.15. It is possible that all the similar proteins containing DAO domain, including glycerol-3-phosphate dehydrogenase-like proteins, followed the incorrect annotation. The misannotation of D-lactate dehydrogenase, D-2-hydroxyglutarate dehydrogenase and as a result other proteins containing FAD_binding_4 and FAD-oxidase_C might stem from the fact that the *E. coli* homolog, encoded by genes in the *glcDEFGB* operon, was initially believed to be a glycolate oxidase belonging to the enzyme class EC 1.1.3.1 [43–45], which was later merged with EC 1.1.3.15.

## Analysing annotation error in the BRENDA database

Biological databases are dynamic by nature and receive regular updates with new experimental information as well as additional proteins from sequenced genomes. We therefore investigated how the annotations to EC 1.1.3.15 changed over time.

In our analysis we compared Pfam domains of sequences annotated to the class in BRENDA 2017.1 and BRENDA 2019.2 (Fig 3C). Over the course of 2.5 years, representing five database versions, the enzyme class grew markedly from 601 sequences to 1659 (excluding redundant and partial sequences). However, the number of sequences containing the canonical FMN-dh domain actually decreased by 11, whereas the newly added sequences are part of clusters containing "non-canonical" protein domains. The most striking rise in sequences in this time period, from 24 to 220 sequences, appeared in the cluster shown by us to contain proteins displaying glycerol-3-phosphate dehydrogenase activity (Pfam domains DAO and Fer2_BFD) *in vitro* as well as that containing the L-2-hydroxyglutarate dehydrogenases (Pfam domain DAO), which rose from 379 to 650 sequences.

This comparison clearly shows that, in the EC 1.1.3.15 enzyme class, the misannotations from old database versions were perpetuated to newly added homologous sequences. Based on the number of sequences lacking the canonical domain architecture alone (absence of the canonical FMN dehydrogenase domain) we estimate that in 2017 at least 78% of sequences in EC 1.1.3.15 are unlikely to catalyse the predicted reaction, while in 2019 this number grew to 87%.

## Investigation of alternative functional predictions for sequences annotated to EC 1.1.3.15 in the BRENDA database

In the BRENDA database the enzymatic function information is extracted manually from scientific literature, but the predicted annotations are imported from external protein databases

[26]. In order to investigate if other annotation methods provide better functional predictions, we scanned the sequences annotated to EC 1.1.3.15 with HAMAP and EggNOG predictors (S1 File). HAMAP [46] classifies and annotates proteins using a collection of expert-curated protein family signatures and annotation rules, while EggNOG [22] is a tool based on fast orthology assignments using precomputed clusters and phylogenies. Both methods provided predictions for only a portion of the input sequences (74% HAMAP, 59% EggNOG), indicating that for some of the sequences there was no evidence for either EC 1.1.3.15, or any other functional prediction. The HAMAP scan provided no annotation that could be directly linked to the S-2-hydroxyacid oxidase activity. Instead, 241 sequences were linked to a function of L-lactate dehydrogenase (MF_011559), and 685 to a function of L-hydroxyglutarate dehydrogenase (ML_00990), which included sequences shown experimentally by us to be active with L-hydroxyglutarate, but also glycerol-3-phosphate (Figs 3 and S8). The EggNOG method assigned 292 sequences with S-2-hydroxyacid oxidase activity (EC 1.1.3.15), two sequences with L-hydroxyglutarate dehydrogenase activity (EC 1.1.99.2), 79 with L-lactate dehydrogenase (EC 1.1.2.3), as well as captured the glycerol-3-phosphate dehydrogenase activity (EC 1.1.5.3) for 54 sequences and D-lactate dehydrogenase activity (1.1.2.4) for 69 sequences. Neither of the methods predicted the experimentally confirmed activities of D4MUV9 (D-lactate dehydrogenase) and A0A077SBA9 (D-2-hydroxyglutarate dehydrogenase). These data show that the use of orthogonal methods of functional annotation can further aid in providing more accurate, if not perfect, functional predictions.

## Exploration of functional annotations in other enzyme classes

In our initial analysis of EC 1.1.3.15 we observed that enzymes from eukaryotes had been disproportionately studied and that a large proportion of sequences annotated to the class shared little similarity with them (Fig 1). We next asked whether EC 1.1.3.15 is a special case, or whether these observations constitute a trend across all of BRENDA. To answer this question we first downloaded all protein sequences from BRENDA 2019.2 and determined which of these have experimental evidence in either BRENDA or SwissProt. We found 30 574 unique identifiers with experimental evidence in SwissProt and 31 287 in BRENDA, only 11 498 of which were overlapping between the two sources. Next, we determined, for each EC class in BRENDA, the degree of identity between each experimentally uncharacterised sequence with the most similar characterised/curated one. To decrease the effect of a large number of similar sequences from repeated sequencing of model organisms we clustered the sequences at 90% using CD-HIT [47] and carried out the subsequent analysis using the ~5.3 million cluster representatives only. As in EC 1.1.3.15 (Fig 1), this global analysis shows that the overwhelming majority of sequences in BRENDA are bacterial (Fig 4A), whereas the majority of experimentally characterised/curated enzymes are eukaryotic (Fig 4B). Furthermore, most enzyme classes have only a small number of characterised/curated enzymes (Fig 4C), indicating that the sequence diversity explored within each EC class is limited.

To analyse the similarity of experimentally uncharacterised sequences to characterised/curated ones we computed, for each EC class, the sequence identity of each cluster representative to the closest characterised enzyme. This analysis is analogous to the one carried out for EC 1.1.3.15 (Fig 1B). The results for all EC classes were aggregated and are presented in Fig 4D, 4E and 4F. In all three superkingdoms the identities roughly follow a normal distribution with a mean below 50% identity. Peaks at 0% represent enzymes for which no characterised homolog is known, and peaks at 100% represent enzymes that have themselves been characterised. We also note peaks around 18% identity, these represent the average pairwise identity of two randomly selected sequences within an EC class (S12 Fig).

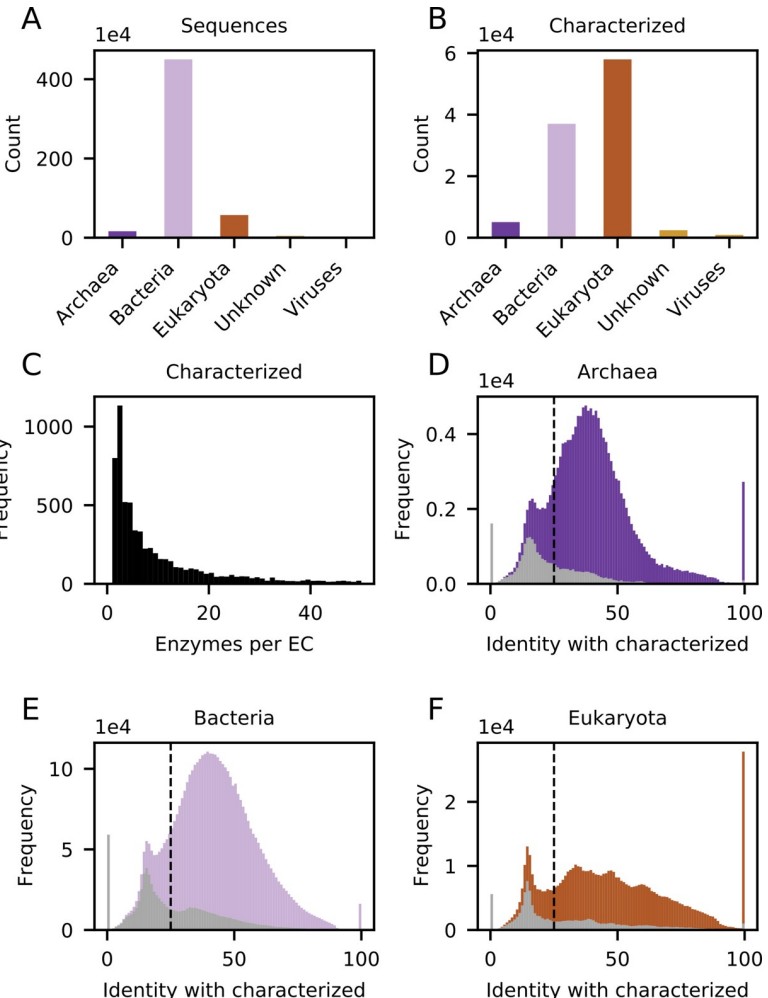

**Fig 4. Exploration of functional annotation throughout all BRENDA enzyme classes. (A)** The total number of representative protein sequences (after clustering at 90% identity) annotated to EC classes in BRENDA, which is approximately 5.3 million. **(B)** The total number of experimentally characterised/curated enzymes. **(C)** Histogram showing the number of characterised/curated enzymes per EC class (bin size of 1). Histograms showing the distribution of sequence identities between all 5.3 million cluster representatives and their closest characterised/curated enzyme for Archaea **(D)**, Bacteria **(E)**, and Eukaryota **(F)** (with a bin size of 1). Proteins which do not have the same Pfam domains as characterised/curated enzymes are coloured in grey.

Strikingly, in each of the superkingdoms almost one fifth of sequences share less than 25% pairwise sequence identity with the closest characterised/curated enzyme–within their own EC class. Such sequences are likely to be incorrectly annotated to a given EC, considering that this is well below the level where function can be confidently transferred between homologous proteins [48,49]. Furthermore, 18% of all sequences, mainly the low-identity ones, are not predicted to have the same Pfam domains as the experimentally characterized enzymes (Fig 4D, 4E and 4F, grey bars), providing further evidence of their likely misannotation. Many such low-homology sequences are annotated even to ostensibly well-characterised enzyme classes with industrially relevant activities (Table 2).

## Discussion

In this study we present experimental investigation of sequence space to explore misannotation in a single enzyme class. By assessing the in vitro catalytic activity of 122 sequences

**Table 2. Overview of annotation to enzyme classes of industrial interest.**

| EC | Name | %id < 25%* | Number of characterised proteins** | Applications [50] |
|---|---|---|---|---|
| 3.1.1.3 | lipase | **54.7** | 141 | detergent, leather processing, pharmaceutical synthesis, degradation of crude oils and plastics |
| 3.1.1.1 | carboxylesterase | **47.6** | 106 | degradation of plastics |
| 3.2.1.4 | cellulase | **30.6** | 191 | pulp and paper processing, detergent |
| 3.2.1.8 | xylanase | **29.9** | 210 | animal feed processing, pulp and paper processing |
| 3.2.1.1 | alpha amylase | **23.9** | 87 | flour adjustment, detergent, leather processing |
| 3.1.1.74 | cutinase | **10.2** | 28 | detergent, degradation of plastics |

* Percentage of sequences in the EC with less than 25% identity to the closest characterised enzyme of the EC

** Proteins listed as characterised in BRENDA database and/or with "experimental evidence at protein level" label in SwissProt

representative of EC 1.1.3.15 in a high-throughput screening experiment we uncovered enzymes which do not display the predicted activity (Figs 2 and 3). Indeed, among the tested enzymes we confirm four alternative catalytic activities which are not compatible with their current annotation. Using sequence homology and protein domain predictions we infer that at least 78% sequences in the enzyme class are possibly misannotated.

In contrast to previous studies investigating annotation errors [24,25], our setup allowed us not only to estimate the error, but also to examine alternative functions of the misannotated sequences. Our experimental approach to the misannotation problem comes with a drawback of limited scope, as we describe in detail only one enzyme class, whereas bioinformatic approaches allow for much broader analysis. However, we argue that our setup is ideal for understudied enzyme classes, and protein families for which experimental evidence is scarce.

The most comprehensive misannotation study so far provided a bioinformatic overview of annotation error in 37 enzyme families in database entries from 2005 [25]. All the analysed families were well-studied and no additional experimental evidence was required to conduct it. Schnoes and coworkers divided the types of misannotation into four categories: "no super-family association", "missing functionally important residues", "superfamily association only", "below trusted HMM cutoff", and showed that the last category is the most prevalent cause of annotation error. This type of error, often called over-annotation, is particularly common in large, multigene families, where enzymes perform similar chemistries on different substrates [51]. In our analysis of EC 1.1.3.15 we also found examples of proteins annotated to the class without functional residues, as well as other members of the superfamily, however, it is the lack of superfamily association that was the main cause of misannotation. In the work by Schnoes et al., which was based on entries to public databases in 2005, only 3% of all sequences were considered misannotated due to the lack of sequence similarity to the gold standard of a superfamily. In our study we show that 15 years later this number is likely much higher now. Similarly to findings described by Schnoes and coworkers [25], we also found a tangible proof of misannotation of enzymes being accumulated, rather than corrected over time (Fig 3C).

Although we did not explore all possible causes of misannotation for all enzyme classes, we show that 17.8% of all sequences annotated in BRENDA share less than 25% sequence identity to the nearest characterised/curated enzyme of the class, and thus are unlikely to perform the predicted function (Fig 4D, 4E and 4F). Similarly, 18.1% of all sequences do not have the same Pfam domains as characterised/curated enzymes from their enzyme class. This is another strong indicator for misannotation, although a portion of this percentage might be explained by missing domains in partially sequenced genes. It is also possible that some of those

sequences indeed perform the predicted activity, however, the records of their experimental characterisation were not registered in BRENDA or SwissProt databases.

In our work we chose to investigate functional annotations to the BRENDA database [38] as it is the premier database linking protein entries with biochemical data, and due to its status as an ELIXIR core data resource (https://elixir-europe.org/platforms/data/core-data-resources). In BRENDA, detailed enzymatic function information is extracted manually from scientific literature, but the predicted annotations are directly imported from UniProt and two of its databases: TrEMBL and Swiss-Prot. Whereas Swiss-Prot annotations are manually curated, and generally highly reliable [25], the TrEMBL section provides automatic and not reviewed annotations, accepting annotations provided during genome submissions, only some of which are corrected by an internal prediction system. Taking into consideration this close link between BRENDA database and UniProt, it is likely that the levels of misannotation to enzyme class shown in our study for the former database are very similar in the latter. It is worth noting, however, that UniProt itself contains a broader description of enzyme function, listing not only an EC number, but also links to other resources predicting protein families, domains, and molecular functions. Resources like InterPro [20], together with its associated databases, attempt to provide more accurate methods for functional annotation, using algorithms relying on protein family signatures or gene ontologies. We show that scanning BRENDA 1.1.3.15 entries with alternatives annotation predictors, HAMAP and EggNOG, provides largely different annotation results. These alternative annotations were in better, although not perfect, agreement with our experimental data than the ones proposed in BRENDA. This highlights the fact that the methods, and as a consequence reliability of functional annotations, vary widely between databases.

With the ever-growing numbers of genomes being sequenced, the gap between experimentally characterised and automatically annotated genes will continue to grow. It is therefore vital that a complete coverage of functional data is available for automated annotation [52]. In our study we characterised four proteins annotated to EC 1.1.3.15 with alternative activities, and in all cases after a literature search we found articles describing homologous proteins with the same activities [39,41,53,54]. Only one article proposed an annotation transfer [39] which resulted in a recent re-annotation of the protein in UniProt (P37339 protein from *E. coli*, L-2-hydroxyglutarate dehydrogenase, EC 1.1.5.13). The remaining proteins are still not recorded in protein databases as being experimentally tested, and thus do not serve as a reliable base for function transfer. Secondary protein databases, such as UniProt or BRENDA, welcome users' corrections, however, it is uncertain to what extent those options are actively used by the community and result in correction of annotations. Initiatives such as COMBREX DB, a database of experimentally validated gene annotations [9], or STRENDA, a guideline of standards for reporting enzymology data [55,56] could help to solve the problem, but only if the whole scientific community adopts these standards. As a response to this issue, the journal *Biochemistry* recently called on authors to include accession IDs for all proteins experimentally characterised in their manuscripts [52], a requirement that should certainly be adopted by other journals. We believe that a structured way of registering proteins characterised in high-throughput experiments should also be developed, and though the depth of protein characterisation in such approaches is limited, they can provide an excellent overview of the substrate scope of a large number of proteins.

Incorrect gene annotations that accumulate over time might have serious consequences for exploration of novelty and understanding fundamentals of biological functions [23]. As shown by us, a number of enzymes with important biological functions were misannotated to the EC 1.1.3.15, including ones taking part in amino acid [39,41], glycerol [53], or lactate [54] metabolism. Even more proteins with functions yet to be discovered might be hidden among the

misannotated sequences. The fields of systems biology [57], metabolic and enzyme engineering [58,59] also rely on accurate annotations, and improved methods for functional annotation are constantly being developed to meet their needs [20,22,60].

## Methods

### EC 1.1.3.15 sequence space analysis

All protein sequences from BRENDA (https://www.brenda-enzymes.org/, version 2017.1) were downloaded and their full UniRep embeddings [37], of 5700 values, were computed. Identical sequences were de-duplicated and multidimensional scaling (MDS) was carried out on the remaining representations using the builtin function in Scikit-learn [61] to decrease the dimensionality of this representation to two, thus allowing visualization as a scatterplot (Fig 1). Taxonomic information for each sequence was obtained by searching for the source organism's name in NCBI Taxonomy resource (https://www.ncbi.nlm.nih.gov/taxonomy). Sequences considered as "characterised" were obtained from UniProtKB/Swiss-Prot (https://www.uniprot.org/) as well as from BRENDA. Specifically, all protein identifiers from UniProtKB/Swiss-Prot (version 2020_02) annotated as belonging to EC 1.1.3.15 and labelled with "Evidence at protein level" were used, as well as those occurring in the "Organism" table of the EC 1.1.3.15 html page in BRENDA (version 2019.1). Pairwise sequence alignments were carried out, using MUSCLE [62], between all 1.1.3.15 sequences. For each sequence the maximum identity to a characterised/curated one was retained (Fig 1B). Pfam protein domain information for each sequence was obtained from UniProtKB. For the domain architectures specified in Fig 1D the arithmetic mean of all pairwise identities was calculated, within each architecture, as well as between architectures.

### Sequence selection for experimental testing

Protein sequences from all EC classes designated as being oxidoreductases acting on hydroxyl groups with oxygen as an electron acceptor (EC 1.1.3.-) were downloaded from BRENDA (version 2017.1) and processed as outlined below, but only sequences from 1.1.3.15 were tested here, the others being reserved for future work. To improve the quality of subsequent alignments, sequences shorter than 200 amino acids (61 total for EC 1.1.3.15) and longer than 580 (31 total for EC 1.1.3.15) were removed, as well as sequences with "X" in them (7 total for EC 1.1.3.15). An all versus all BLAST was carried out using plastp from BLAST+ [63] with standard settings, followed by clustering using the MCL algorithm [64] with standard settings, except for the inflation parameter -I, which was set to 1.4. This resulted in 17 clusters. A multiple-sequence alignment was created for each cluster using MUSCLE [62]. The Shannon entropy—a metric quantifying the degree of conservation at each position—was used to select a diverse and informative set of sequences for testing. The metric was calculated for each multiple sequence alignment and sequences were then iteratively selected such that each newly chosen one maximally increased the information gain; they were chosen to maximize the mutual information explained within each alignment. This iterative sequence selection was continued until 85% of the information in each cluster had been explained.

### Cloning, expression of sequences and protein purification

Generated sequences were synthesised, cloned into the pET21a vector and sequenced-verified by Twist Bioscience. Between the gene sequence and vector backbone, a C-terminal linker was added (AAALEHHHH), which in combination with six histidines from an expression vector resulted in a deca-His-tag for improved protein purification. The 122 constructs used in this

work were deposited to Addgene (https://www.addgene.org/) with IDs 163180–163301. High throughput expression, lysis and, when necessary, purification was carried according to the published protocol [65]. Briefly, expression was carried in *E. coli* BL21(DE3) cells, in 96-well deep well plates, in 1 ml autoinduction TB (Foremedium). After cell lysis, cells were spun down and supernatants analysed by SDS-PAGE followed by Coomassie staining (InstantBlues, Expedeon). Each sequence was expressed three times; a sequence was scored as soluble when the corresponding band was present on a gel in at least two expressions. The soluble fraction of the lysate was used for the screen of S-2-hydroxyacid oxidase activity, whereas affinity-purified proteins were used for the dehydrogenase activity screen and determination of kinetic parameters.

## Activity assays

To screen for S-2-hydroxyacid oxidase activity, lysates of soluble proteins were assayed in the Amplex Red hydrogen peroxide detection assay (Fisher Scientific) with a selection of 2-hydroxyacids: glycolate, L-lactate, DL-2-hydroxyoctanoate, DL-2-hydroxyoctadecanoate, DL-mandelate, L-2-hydroxyglutarate. Each protein was assayed three times and was considered a hit if it was scored as soluble and active at least twice. 1 µl of soluble fraction of the lysate after protein expression was added to a reaction mixture containing 20 mM HEPES pH 7.4, 50 µM Amplex Red (Fisher Scientific), 0.1 U/ml horseradish peroxidase (HRP) and 1 mM of an appropriate substrate. Final reaction volume was 20 µl, and the assay was performed in black 384-well low volume plates (Greiner). After 30 minutes of incubation in the dark, the endpoint measurements were performed with an excitation filter of 544 nm and emission filter of 590 nm in a BMG Labtech FLUOstar Omega microplate reader. Each reaction was performed in triplicate. Values for non-specific activity in the absence of substrate were subtracted from experimental measurements. *E. coli* lysate from cells expressing BSA protein was used as a control to establish a limit of detection of the assay ($mean_{BSA} + 4^*SD_{BSA}$).

For the dehydrogenase activity screening and kinetic characterisation, proteins were purified by affinity purification, and assayed with a range of substrates and electron acceptors. Purified protein in the volume of 1 µl was added to a reaction mixture containing 20 mM HEPES pH 7.4, 2 mM of substrate and electron acceptor. L-lactate (cytochrome) dehydrogenase activity was tested with 0.1 mM cytochrome c as electron acceptor. Glycerol-3-phosphate dehydrogenase activity was tested with the following electron acceptors: 0.2 mM DCPIP + 3 mM PMS, 50 µM Amplex Red + 0.1U/ml HRP, 1mM NAD, 1mM NADP. 2-hydroxyacid dehydrogenase activity was tested with all the above electron acceptors, with the addition of 0.15 mM cytochrome c. Activity was measured in triplicate every 30 seconds over 15 minutes at 340 nm in the case of NAD and NADP, at 600 nm in the case of DCPIP/ PMS, at 550 nm in the case of cytochrome c, and with excitation/emission filter of 544 nm/ 590 nm in the case of Amplex Red/HRP. Unspecific reduction of electron acceptor was monitored in controls lacking substrate, and the values were subtracted from experimental measurements.

The kinetic values for four chosen proteins were determined at 25˚C with DCPIP + PMS as electron acceptor and a varied range of substrate concentrations. Protein concentrations used for the assays were: 60 nM D4MUV9, 50 nM A0A077SBA9 with D-2-hydroxyglutarate, 1.3 µM A0A077SBA9 with D-malate, 25 nM S2DJ52, 660 nM A0A0R3K2G2. Activities were calculated using the extinction coefficient of DCPIP at 600 nm (20.7 mM$^{-1}$cm$^{-1}$).

Comparison of DCPIP and AR reaction rates was carried for the four characterised proteins. Reactions rates were performed for both electron acceptors, using concentration values of proteins and substrates as listed above.

### EC 1.1.3.15 annotation over time

All EC 1.1.3.15 sequences were downloaded from two BRENDA versions, differing by 2.5 years in their publication (versions 2017.1 and 2019.2). Identical sequences in each database version were de-duplicated, resulting in 1058 sequences from 2017.1 and 1659 sequences from 2019.2. Pfam domains for these sequences were obtained by querying UniProt using the protein identifiers, and mining the resulting page for domain data. The frequency of each domain was subsequently computed.

### Exploration of alternative annotations

Sequences listed in the file "1_1_3_15_BRENDA_sequences_filtered_2017_1.fasta" were uploaded for the scans by HAMAP (https://hamap.expasy.org/hamap_scan.html) and egg-NOG-mapper v2 (http://eggnog-mapper.embl.de/). Xlsx results files from the scans were downloaded.

### Exploration of annotation quality throughout enzyme classes

A list of UniProt identifiers for enzymes considered "characterised" was compiled from SwissProt and BRENDA as described in the first Methods section. Protein sequences from all EC classes were downloaded from BRENDA (version 2019.2). Within each EC class, sequences were clustered to 90% identity using CD-HIT [47] with standard settings and a word size of 5. Cluster representatives were retained for subsequent analysis. Since the clustering had resulted in some "characterised" sequences to be removed (they were not cluster representatives) these were added back. For every cluster representative within each EC class the sequence identity to the closest characterised/curated sequence (within that class) was computed. First, an alignment-free measure of similarity was obtained using the alfpy package [66] by computing count-based k-tuples with word size of 3 and Normalised Google Similarity [67] as a distance measure (S11 Fig). For each uncharacterised-characterised pair with highest k-tuple-based similarity, pairwise sequence alignments were created using MUSCLE and the sequence identities calculated. These are the identities reported. The superkingdom of the source organism was obtained for each organism, firstly by matching the organism name with the NCBI-Taxonomy database, and secondly by querying UniProt using the protein identifiers. Pfam (release 33.1) domain information was obtained from the "Pfam-A.full.uniprot" file provided at the FTP site (ftp://ftp.ebi.ac.uk/pub/databases/Pfam/). Two proteins were scored as having the same Pfam domains only in cases where all domains matched, but disregarding their order.

### Software

Scripts used to analyse data and generate manuscript figures are available as a GitHub (https://github.com) repository: https://github.com/EngqvistLab/analyze_1.1.3.15. All software packages, with their versions, are specified in a Miniconda (https://docs.conda.io) environment file in that repository. Briefly, analysis was carried out using the Python programming language version 3.7 (http://www.python.org), using the following packages: Biopython version 1.76 [68], Pandas version 1.0.1, Numpy version 1.18.1 [69], Matplotlib version 3.1.3 [70], Scikit-learn version 0.20.0 [61], TensorFlow version 1.15.0 (https://www.tensorflow.org/), Networkx version 2.5 (https://networkx.org/), Jupyter version 1.0.0 (https://jupyter.org/), Alfpy version 1.0.6 [66], BeautifulSoup4 version 4.9.3 (https://www.crummy.com/software/BeautifulSoup/). Additionally, the following standalone software was used: MUSCLE version 3.8.1551 [62], CD-HIT version 4.8.1 [47], MCL version 14.137 [64], BLAST+ version 2.5.0 [63], and UniRep [37].

## Supporting information

**S1 Fig. Schematic representation of the reaction catalysed by S-2-hydroxyacid oxidases (EC 1.1.3.15).**
(TIFF)

**S2 Fig. A hexbin plot indicating Euclidean pairwise distances between the 1058 proteins annotated to EC 1.1.3.15.** Clustering along the diagonal indicates that the multidimensional scaling (MDS) dimensionality reduction faithfully represents pairwise distances of the UniRep representations of these sequences. The total number of pairwise distances is indicated, corresponding to half of the distance matrix, without the diagonal.
(TIFF)

**S3 Fig. Identity of sequences annotated as EC 1.1.3.15 to the closest characterised S-2-hydroxyacid oxidase.**
(TIFF)

**S4 Fig. Distribution of the insoluble, active and inactive screened proteins throughout the sequence space (left panel) and superkingdoms (right panel).**
(TIFF)

**S5 Fig. S-2-hydroxyacid substrates used for the screening of EC 1.1.3.15 sequence space.** The donor group is marked in red.
(TIFF)

**S6 Fig. Multiple sequence alignment of previously characterised representatives of the FMN-dependent 2-hydroxyacid oxidase/dehydrogenase family and proteins characterised in the study.** Conserved residues around the active site are circled in red. Sequence of predicted heme binding domain is highlighted in green, the elongated loop 4 is highlighted in blue. MSA performed in PROMALS3D (1) and visualised with Multiple Align Show (https://bioinformatics.org/sms/).
(TIFF)

**S7 Fig. Cytochrome c reduction assay of putative flavocytochrome b2 proteins.** Increase of signal at the wavelength of 550 nm indicates reduction of cytochrome c and protein activity.
(TIFF)

**S8 Fig. Exploration of alternative activities of selected proteins.** Presence of activity is marked with a dark purple square. (A) glycerol-3-phosphate dehydrogenase activity screen (B) 2-hydroxyglutarate dehydrogenase activity screen.
(TIFF)

**S9 Fig. Comparison of sensitivity of Amplex Red and 2,6-dichlorophenolindophenol (DCPIP)-based assays.** (A) Standard curves of resorufin, a product of Amplex Red-based assay (upper panel) and DCPIP (lower panel). Indicated by asterisk are concentrations of detection limit, as calculated by Anova single factor test (0.76 nM resorufin, 1.56 μM DCPIP). (B) Reaction rates of selected enzymes with the two electron acceptors, normalised to the reaction rate with DCPIP. Error bars in all figures represent standard deviation of the data obtained with three replicates.
(TIFF)

**S10 Fig. Characterisation of proteins with activities alternative to 1.1.3.15.** (A) SDS-PAGE gel of purified proteins chosen for kinetic characterisation. (B) Kinetic curves of the

characterised enzymes. Error bars show standard error of three replicates.
(TIFF)

**S11 Fig. Test to find best k-tuple algorithm settings.** Using 400 randomly selected protein sequences all pairwise distances were calculated using different word size and distance measures. These distances were compared to distances computed using pairwise alignments. Appropriate k-tuple settings will cause points to lie on a diagonal, thus showing a high degree of correlation with the alignment-based values. Spearman's rho and p-value is indicated for each plot.
(TIFF)

**S12 Fig. Average similarity between 400 randomly selected sequences from EC 1.1.3.15 (left panel), using k-tuple scores and pairwise alignments (right panel).** The k-tuple score was computed using a word size of 3 and google as a distance measure. The mean alignment-based identity is 18%. The total number of pairwise similarities is indicated, corresponding to half of the identity matrix, without the diagonal.
(TIFF)

**S1 File. Results of a functional prediction scan of sequences annotated to EC 1.1.3.15 in BRENDA 2017.1 using HAMAP and EggNOG servers.**
(XLSX)

## Acknowledgments

The authors express their gratitude to Martin Lercher and Jan Zrimec for critical reading of the manuscript. The computations were performed on resources provided by Chalmers e-Commons (C3SE).

## Author Contributions

**Conceptualization:** Elzbieta Rembeza, Martin K. M. Engqvist.

**Data curation:** Elzbieta Rembeza.

**Formal analysis:** Elzbieta Rembeza, Martin K. M. Engqvist.

**Investigation:** Elzbieta Rembeza.

**Methodology:** Elzbieta Rembeza, Martin K. M. Engqvist.

**Project administration:** Martin K. M. Engqvist.

**Software:** Martin K. M. Engqvist.

**Supervision:** Martin K. M. Engqvist.

**Visualization:** Elzbieta Rembeza, Martin K. M. Engqvist.

**Writing – original draft:** Elzbieta Rembeza, Martin K. M. Engqvist.

**Writing – review & editing:** Elzbieta Rembeza, Martin K. M. Engqvist.

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
