## [Decision Letter · Decision Letter 0]

4 Jun 2021

Dear Dr. Engqvist,

Thank you very much for submitting your manuscript "Experimental and computational investigation of enzyme functional annotations reveals extensive annotation error" for consideration at PLOS Computational Biology.

As with all papers reviewed by the journal, your manuscript was reviewed by members of the editorial board and by several independent reviewers. In light of the reviews (below this email), we would like to invite the resubmission of a significantly-revised version that takes into account the reviewers' comments.

While I ask that you respond carefully to comments made by all reviewers, I would like you to pay particular attention to criticism and suggestions from reviewers 1 and 4 who call for a more in depth and balanced discussion of existing in silico annotation, within and outside of the BRENDA database and, in particular, for a more careful consideration of the different sources/criteria of/for these annotations and thus of their different level of expected reliability. 

We cannot make any decision about publication until we have seen the revised manuscript and your response to the reviewers' comments. Your revised manuscript is also likely to be sent to reviewers for further evaluation.

Sincerely,

Marco Punta

Associate Editor

PLOS Computational Biology

Arne Elofsson

Deputy Editor

PLOS Computational Biology

Reviewer's Responses to Questions

**Comments to the Authors:**

Reviewer #1: BACKGROUND

This paper addresses an important question that concerns many scientists who have the responsibility of maintaining databases of annotated proteins. How can a fair and complete description of the function of all of the proteins in the database be provided when so few of the proteins have been experimentally characterised?

GENERAL COMMENTS

The paper includes a section on the synthetic construction of a number of genes for uncharacterised proteins reported in the BRENDA database to have activity as EC 1.1.3.15. I found this section well argued and the authors showed an awareness of the limitations of the experimental techniques they were using.

The other half of the paper discusses what our response should be, given that the experimental data gathered by the authors shows that many of the uncharacterised proteins that they examined had a different activity from the one annotated in the BRENDA database.

I have several misgivings about the line of argument followed in this section. I found the description of the process of experimental characterisation of proteins, and the consequent opportunity to propagate annotation to uncharacterised proteins lacked an understanding of the true state of play currently in the annotation of proteins. There was trenchant criticism of the annotations in one database (BRENDA) leading to a suggestion that all automated annotation of protein function is essentially flawed. This is not a correct assessment of the current state of play, and undermines the cooperative relationship that exists between those who provide experimental evidence of protein function, and those who use bioinformatic approaches to propagate this information to uncharacterised proteins.

The authors seem unaware of the wider discussion about how to propagate annotation by sequence similarity. See, for instance, Pearson WR, Protein Function Prediction: Problems and Pitfalls. Curr. Protoc. Bioinform. 51:4.12.1-4.12.8. doi: 10.1002/0471250953.bi0412s51.

The authors do compare their data with domain annotation provided by Pfam, but there is no mention of InterPro or the InterPro member databases (including Pfam) which try to generate models that capture protein families and not just domains. For instance, the HAMAP signature MF_00990 (https://hamap.expasy.org/rule/MF_00990) which is associated with EC 1.1.5.13 and which occurs in 210 of the 1414 records found in the supplementary file 1_1_3_15_BRENDA_sequences_filtered_2017_1.fasta. Agreed, there is only one characterised entry for MF_00990, but this family signature shows there are alternative annotations available for a good proportion of the BRENDA annotated EC 1.1.3.15 proteins.

Also the Functional Families of the CATH database have representatives which touch the records the authors are looking at. Notably CATH 3.20.20.70, functional family 63 (http://www.cathdb.info/version/v4_3_0/superfamily/3.20.20.70/funfam/63/ec). This functional family includes 10 UniProt reviewed entries with EC 1.1.3.15 (all plant species) and 202 UniProt unreviewed entries which are also all plants. So there are routes to EC 1.1.3.15 prediction which respect the taxonomic imbalance of the experimentally characterised proteins.

Building on from InterPro member databases that generate family signatures, there is the annotation effort of UniProt, which includes both curated rules (UniRule) and a fully automated prediction system (ARBA). The significance of this work is that it does not attempt to annotate everything, and limits itself to propagating annotation where there is consistent annotation of the characterised proteins, and a suitable sequence model that can be used to identify proteins of similar function.

Out of the 1411 proteins listed in the Supplementary Material file 1_1_3_15_BRENDA_sequences_filtered_2017_1.fasta, and which map to entries in UniProt release 2021_02, UniRule annotates 12 plant proteins with EC 1.1.3.15 through ARBA00013087 (https://www.uniprot.org/arba/ARBA00013087), and 210 proteins from the Enterobacterales with EC 1.1.5.13 through MF_00990 (https://www.uniprot.org/unirule/UR000165710).

It is true that BRENDA follows a policy of providing maximum coverage of annotation, as is made clear in the protocol for gathering annotation shown in Fig 1 of Quester and Schomburg BMC Bioinformatics 2011, 12:376 (http://www.biomedcentral.com/1471-2105/12/376). Also, the TrEMBL section of UniProt uncritically reports enzyme activities provided during submission of genome sequences until these are overwritten by a prediction system. These are indeed policies that the authors can validly criticise on the basis of their data.

However, the best current approaches to annotation in the major protein databases do not attempt to annotate everything, but use a mixture of protein family signatures and taxonomic constraints to provide as good annotation as can be provided based on the current experimental data. This annotation clearly disagrees with both the extent and content of the annotation provided by BRENDA, showing there is a problem, even without the experimental data the paper provides.

Focusing only on the shortcomings of BRENDA annotation and ignoring these other annotation approaches results in an unbalanced and misleading discussion. There is also a missed opportunity here: to target the selection of genes to synthesise to include representatives which are touched by the family signatures mentioned above, and thereby assess the reliability of these signatures.

SPECIFIC REQUIREMENTS

For me to be able to recommend this paper for publication, the discussion of the significance of the experimental data for BRENDA and other protein databases needs a good deal of revision.

1. The authors should not claim, either in the title or in the text, that their paper demonstrates a general failure of enzyme functional annotation in protein databases. They should also not claim that this is the first time that a significant mismatch has been shown between database annotation and subsequent experimental data. This has previously been thoroughly covered in a paper the authors quote (Schnoes et al., Plos Computational Biology, 11 Dec 2009, 5(12), DOI: 10.1371/journal.pcbi.1000605). The current paper is a continuation of this, focusing on BRENDA instead of KEGG.

2. The authors need to make it clear in the text that they understand that experimental evidence is for the foreseeable future going to remain in very short supply, and that there is a valid role for automated annotation of protein function. On page 24, paragraph 1, the authors slip into the error of suggesting that the only true answer to reliable annotation of proteins is for much greater prominence to be given to experimental evidences (and by implication, a much greater quantity of these). This is naive and shows a lack of appreciation of the size of the task of protein annotation, and the mutually reinforcing roles played by experimentation and prediction in maximising the number of well annotated proteins.

3. The authors need to show that they can distinguish the different qualities of annotation (arising from different approaches to automated annotation) that are found in protein databases, as described above.

4. The relationship between the authors' clustering method and multiple sequence alignment methods needs to be more clearly discussed. Curiously the data that the authors could be using to point to a better way forward in protein annotation is in the paper itself. The authors state on Page 6 line 14 that they wish to avoid grouping proteins based on multiple sequence alignment and use an approach from protein engineering. Having done this they then display the level of agreement between the clustering they have used and the presence of several different Pfam domains. For PF01070 and PF01266 the clustering the authors have used arrives at the same place that is provided by multiple sequence alignment and the building of sequence models. So there needs to be a better discussion of the two approaches to sequence clustering instead of the implied assertion that alignment-free comparisons are inherantly superior.

As a side note, it seems clear from the method paper quoted by the authors (https://doi.org/10.1038/s41592-019-0598-1), that the clustering approach used is likely to be too computationally expensive to be implemented in the major protein databases. So it provides a useful way of checking the results achieved by methods based on multiple sequence alignment, but is not an approach that could replace them.

5. The length of the discussion on annotation in BRENDA over time should be significantly reduced, as this point has already been well made for KEGG previously in the paper by Schnoes et al.

Finally, I did appreciate the provision of a full set of supplementary data which was very helpful for carrying out this review.

PROOF READING

These are suggestions for the authors to consider. Some are in sections of the manuscript that I am asking to be extensively revised, and should not be taken to indicate that I agree with the text.

Page 3 line 1, Change to: 'utilization of functional gene diversity' (delete 'the')

Page 3 line 5, Change to: 'at least 78% of the sequences' (add 'of the')

Page 4 line 12, 'initiatives were undertaken' change to 'initiatives have been undertaken'

Page 4 line 12, Change to: 'bring together computational and' (delete 'the')

Page 5 line 2, 'estimated the annotation error between' change to 'estimated the annotation error to be between'

Page 5 line 3, 'depending on a protein' change to 'depending on the protein'

Page 5 line 14, 'BRENDA DB' change to 'the BRENDA database' (For consistency with how BRENDA is referenced elsewhere.)

Page 6 line 16, '17 of these' Most journals don't allow you to start a sentence with a number in digits.

Page 6 line 17, Change to: 'evidence at the protein level' (add 'the')

Page 7 line 15, Change to: 'identity to previously characterised' (delete 'the')

Page 8 line 3, Change to: 'Pfam domain architecture' (Delete 'Predicted'. I suggest this is redundant as Pfam is a prediction in its nature.)

Page 10 line 8, Change to: 'all the members of the FMN-dependent' (add 'the')

Page 10 line 10, 'Fig 2C)' (Missing opening bracket.)

Page 10 line 19, Change to: 'Amplex Red assay, the four' (Add a comma.)

Page 13 line 13, Change to: 'the Pfam [30] domains' (Delete 'predicted'. I suggest this is redundant as Pfam is a prediction in its nature.)

Page 15 line 1, Change 'proved to show' to 'showed'

Page 16 line 2, Change 'indicates' to 'indicating'

Page 16 line 3, Change to: 'marked with squares; for proteins' (A semicolon instead of a comma, or a new sentence.)

Page 16 line 12, Change to: 'Comparison of Pfam domains' (Delete 'predicted'. I suggest this is redundant as Pfam is a prediction in its nature.)

Page 16 Table 1, Three decimal places is very unlikely to be justified by the data. I suggest two.

Page 17 line 4, Change to: 'we compared Pfam domains' (Delete 'predicted'. I suggest this is redundant as Pfam is a prediction in its nature.)

Page 21 Table legend, Change 'BRENDA DB' to 'the BRENDA database' (For consistency with how BRENDA is referenced elsewhere.)

Page 22 line 3, Change to: 'In contrast to previous studies' (delete 'the')

Page 24 line 10, Change to: 'called on authors' (add 'on')

Page 24 last line, Change 'will be of much higher standards.' to 'will be of a much higher standard.'

Page 26 line 6, Change to: 'the source organism's name in the NCBI' (Add apostrophe and 'the'.)

Page 27 line 10-13, 'The Shannon ... each cluster.' The meaning of this sentence is unclear to me.

Page 28 line 2, Change to: 'carried out' (Add 'out'.)

Page 28 line 5, Change to: ' was expressed three times; a sequence' (A semicolon instead of a comma, or a new sentence.)

Page 28 line 6, Change to: 'The soluble fraction of' (Add 'The'.)

Page 28 line 8, Change to: ' activity screen and determination of kinetic parameters.' (Rephrased)

Page 28 line 10, Change to: 'To screen for S-2-hydroxyacid' (Delete 'the')

Page 28 line 16, Is HRP a permitted abbreviation or does horse radish peroxidase need to be given somewhere?

Page 28 line 17, Change to: 'volume was 20 ul, and the assay' (Add 'and')

Page 28 line 20, Change 'triplicates' to 'triplicate'

Page 28 line 21, Change to: 'Values for non-specific activity in the absence of substrate were subtracted from experimental measurements.' (Rephrased)

Page 29 line 4, '1 ul of purified' Most journals don't allow you to start a sentence with a number in digits.

Page 29 line 6, Change to: 'L-Lactate' (Capitalisation of word.)

Page 29 line 8, Change to: 'tested with the following' (Add 'the'.)

Page 29 line 9, Change to: '2-Hydroxyacid' (Capitalisation of word.)

Page 29 line 11, Change 'triplicates' to 'triplicate'

Page 29 line 11-13, Change 'in case of' to 'in the case of' (Four changes.)

Page 29 line 14, Change to: 'monitored in controls lacking substrate, and the values were subtracted from experimental measurements.' (Rephrased)

Page 29 line 18, Change to: 'used for the assays were:' (Add 'were'.)

Page 29 line 22, 'Reaction rates ... electron acceptors' (This sentence is unclear to me.)

Page 30 line 9, 'Change to: ' Within each EC class, sequences' (Add comma,)

Page 30 line 17 Change to: 'highest k-tuple-based similarity, pairwise sequence' (Add comma.)

Reviewer #2: This article addresses a very important question which, despite its ancient origin, remains timely. While the topic of pervasive annotation errors is very general, the authors chose to illustrate the situation with a specific case, that of S-2-hydroxyacid oxidases. This work is therefore of interest for metabolic engineering studies. However a biological justification of this choice would have improved readers' interest in this article.

Overall, the work provides an interesting and well documented study on the general problem posed by percolation of wrong annotations in public open databases. While the authors rightly point out the dangerous situation we are facing, this is not novel knowledge. As a matter of fact, over the years, several works focused on misannotations. This should probably be further emphasized in the introduction of the article as this could help readers to uncover other, highly relevant, related approaches. It would benefit the work to provide the readers with references to earlier attempts to tackle the question, in particular PMID 12490449, for example. Reference to works such as PMID 17708678, 28525546 in complement to 29806194, that was cited by the authors, would also help readers to understand how attempts were made to remedy this situation, with not much success, unfortunately. As the authors remarked, there is a need to couple in silico analyses with predictions and experimental attempts to validate the predictions. Again, a biological justification of their choice would be welcome.

As a matter of fact, in a way quite similar to that proposed by the authors in the present paper, Risler and co-workers, twenty years ago, developed a work that associated an original in silico approach with experiments meant to identify explicitly the enzyme activities predicted in their work. This bioinformatics/experimental work focused on the differences between arginases and agmatinases. It validated experimentally the predictions. This early attempt seems highly reminiscent of the present one (PMID 10931887).

This early work pointed out that it can be expected that methods should differ when looking into activities that correspond to sequences that diverged recently or slowly (implying that amino acid changes are relevant, such as those involved in catalysis PMID: 31733177), or into sequences that diverged a very long time ago or rapidly (such as in virus evolution), where it is likely that only the global 3D structure is conserved, with only catalytic residues preserved (implying that 3D features found in insertions/deletions would be relevant). In this case, constructing phylogenies based on indel trees, might be relevant. This feature has been recently used to characterize relevant traits of the SARS-COV-2 descent, for example (PMID 33125064). The very specific case of 2-hydroxyglutarate oxidation, which is so important in a variety of regulatory or metabolic contexts would probably benefit from this approach in another work.

This observations makes this reviewer regret that there is so little biologically relevant information discussed in this paper. After all, annotation is meant to help investigators to progress in their understanding of biological functions, and some comments about the consequences of wrong annotations associated to the class of enzymes studied in the present work would have been more than welcome (and would have increased considerably the visibility of the work...)

Reviewer #3: In the manuscript by Rembeza and Engqvist the authors assess 122 representative sequences of those annotated using E.C. class as S-2-hydroxyacid oxidases and find, by inference to related sequences, that 78% of the class is misannotated. The extension of the analysis computationally using the BRENDA database shows a high percentage of miss-annotation within a class to enzymes sharing no similarity or domain architecture. These findings will be useful to the general community and those specializing in the areas of enzyme structure and function. The work is technically well performed and well presented. The authors may wish to consider the following points (page numbers from PDF for review and in order of appearance).

Page: 8- The global initiatives cited should probably include the enzyme function initiative summarized in Biochemistry. 2011 Nov 22;50(46):9950-62. doi: 10.1021/bi201312u

Page: 9- EC number is assigned by activity not by sequence. Here EC 1.1.3.15 was chosen for study, but the first section in Results immediately point out the dissimilarity in sequence and fold. However, this might actually be expected using EC versus CATH or SCOP database to start. To make the work more accessible to the general reader, the authors should include in the beginning of results or the introduction, a brief description of the EC classification system and point out that the structure and sequence would not necessarily be expected to be the same within an EC class, but that the function should be the same. However, enzymes annotated using sequence identity may have a similar fold at best, but may indeed have different functions.

Page: 11- The authors state “Most sequences have little similarity with the characterised ones; 79% of sequences annotated as 1.1.3.15 share less than 25% sequence identity with the closest biochemically characterized sequence (Fig. 1B, Fig. S3). Furthermore, only 22.5% of the 1058 sequences are predicted to contain the FMN-dependent dehydrogenase domain (FMN_dh, PF01070) which is canonical for

known 2-hydroxy acid oxidases (Fig. 1C). Can the authors posit at all how the annotations were originally made and what the mis-step was in that assignment? Was it merely using too low a threshold for sequence identity and then, as the authors later find, misannotations from old database versions perpetuated to newly added homologous sequences?

Among the 24 proteins with the FMN-dh domain, for the proteins that were inactive, were studies performed to ensure that they were folded (ie. by CD or light scattering)? If not perhaps it should be described that the proteins were either misfolded or inactive.

Table 1 - please give Km in M not mM values- Vmax is not useful- instead please give kcat/Km in M-1 s-1 so the reader does not need to calculate

Page: 18- The authors state “Three of the four enzymes (D4MUV9, A0A077SBA9, S2DJ52) had substrate

affinities in the micromolar range and high catalytic rates, strengthening the possibility that these

may be the natural substrates. As noted for Table 1 the column with Vmax should be replaced with kcat/Km which should be used to discuss enzyme efficiency. I would say the vales here should not be described as high (the greatest being 7.8 x 10^4 M-1 s-1) but does approach that typically used as a cutoff for a physiologically relevant substrate ~ 1 x 10^5 M-1 s-1 (reference 4 in manuscript).

Minor changes/typos

Page: 10- The authors state “17 of these sequences are characterised enzymes: either listed in BRENDA [17] as experimentally tested or in SwissProt [1] as having experimental evidence at protein level.” Do not start sentence with a number.

Page: 13- “…glycolate, lactate, 2-hydroxyoctanoate, 2-hydroxydecanoate, mandelate, 2-hydroxyglutarate" should read "... mandelate, and 2-hydroxyglutarate

Page: 14- "Indeed, the B8MKR3 protein displayed the cytochrome b2 L-lactate dehydrogenase activity" should read "... protein displayed cytochrome b2 L-lactate dehydrogenase activity"

Page: 26- "In the work by Schnoes et al., based on entries to public databases in 2006, only 3 % of all sequences were considered misannotated due to the lack of similarity to the golden standard of a superfamily, in our study we show that this number is likely much higher now." This is an awkward sentence although meaning is clear- first it is the gold not golden standard. Second this should be broken into two sentences.

Page: 28- “Only one article postulated for annotation transfer”- I think this should read "Only one article proposed an annotation transfer"

Page 28- “high-throughput experiments should also be developed, as though the depth of protein

characterisation in such approaches is limited" should read "...also be developed, and though the depth of protein..."

Reviewer #4: The authors present a very careful analysis of the functional assignments of enzymes as available from public databases. Taking S-2-hydroxyacid oxidases (EC 1.1.3.15) as an example they analyzed the sequences of more than 1000 proteins with a predicted activity of this class. Only 17 examples of these had an experimental characterization, 14 of them of eukaryotic origin.

They found that almost 80% off the sequences assigned to this EC number have less than 25% sequence identity to the closest experimentally proven one, and only 22.5% are predicted to contain the FMN-dependent dehydrogenase domain. In fact five different Pfam domains were found among the sequences. They took 122 sequences to try an experimental check of their enzyme activity. Out of these 65% could be expressed in a soluble form and experimentally tested for the S-2-hydroxy acid oxidase activity with six different substrates.

The expressed proteins containing the FMN_dh were checked for the enzyme activity, only partially with success. Out of the 41 expressed protein that do not have the FMN_dh domain most did not show an enzyme activity corresponding to the EC number 1.1.3.15, some of them displayed a related dehydrogenase activity, e.g. D-lactate dehydrogenase activity. Again, some of them were further purifiied for the determination of kinetic constants.

All in all the authors came to the result, that those proteins that do not contain the canonical FMN-dh domain probably have other catalytic activities and their annotation as S-2-hydroxyacid oxidases (EC 1.1.3.15) is probably not correct. The authors have found that these probably misannotated proteins represent almost 80% of the sequences downloadable from Uniprot or BRENDA.

In a final chapter the authors apply the results from their specific analysis to all annotated enzyme sequences downloadable from UNIPROT/BRENDA and find that the large majority of sequences show a sequence identity with the closest experimentally characterized representative of their EC-Class of more than 30%. The authors point out, that on the other hand 20% of the sequences share less than 25% pairwise sequence identity with the closest characterized enzyme in their own EC-class.

Overall this is a very good and important paper, combining theoretical analyses with a large number of experimental data. In general it is well written. Nevertheless there are a number of misunderstandings/errors in interpretation that have to be corrected before publication. In particular the general conclusions in the discussion are not all justified.

The following modifications and clarifications are absolutely essential:

1) As becomes obvious from the BRENDA publications the function assignment for the downloadable sequences is directly imported from UNIPROT. UNIPROT has two datasets: the SWISSPROT sequences (presently 23 sequences for 1.1.3.15), and the TREMBL dataset (presently 3590 sequences). On the interactive UNIPROT pages they are named “Reviewed” or “Unreviewed” and in the BRENDA downloadable data their source is given an SWISSPROT or TREMBL. Whereas the SWISSPROT data are manually checked by the UNIPROT scientists and should be highly reliable, the source of the functional assignment of TREMBL sequences is described in the SWISSPROT/TREMBL guide by the following “Automated annotation of the highest currently available quality is integrated to TrEMBL entries.”

Only for those enzymes where experimental data are available in BRENDA the EC numbers were manually checked and sometimes corrected by the BRENDA team. For the EC class 1.1.3.15 these are presently 15 sequences.

So, these annotation errors appear as errors in the BRENDA database but in fact are really errors in UNIPROT. This should be obvious because the source of the sequence/annotation is given in the downloadable file.

2) The label “Evidence on protein level” for UNIPROT sequences does not mean that there is experimental evidence for the function but only for the existence of a protein. Again, the UNIPROT guide says:

“The value 'Experimental evidence at protein level' indicates that there is clear experimental evidence for the existence of the protein. The criteria include partial or complete Edman sequencing, clear identification by mass spectrometry, X-ray or NMR structure, good quality protein-protein interaction or detection of the protein by antibodies.”

For the description of the protein function there are the so-called evidence codes in UNIPROT. One can find an explanation of the different codes here: http://www.uniprot.org/help/evidences. In short, 255 is added by automatic procedures, 250 is by similarity, and 269 is experimental evidence.

3) Looking at Figure 1 B one really does not get the impression that 80% of the sequences have less than 25% identity to the closest experimentally tested one. One would guess from the figure that this is only true for about 20%! This must be checked.

4) In the general discussion the authors say “Strikingly, in each of the superkingdoms almost one fifth of sequence share less than 25% pairwise identity with the closes characterized enzyme.” This statement as it stands gives the expression that this is unexpected - which is not really true.

An EC-class is assigned to a protein based on its enzymatic function, i.e. the catalyzed reactions and their substrate specificity if there are clear differences observed.

For example, for EC 1.1.3.15 the IUBMB description says:

“A flavoprotein (FMN). Exists as two major isoenzymes; the A form preferentially oxidizes short-chain aliphatic hydroxy acids, and was previously listed as EC 1.1.3.1, glycolate oxidase; the B form preferentially oxidizes long-chain and aromatic hydroxy acids. The rat isoenzyme B also acts as EC 1.4.3.2, L-amino-acid oxidase. “

It could be, and this is indeed very often the case, that there are several sequence families that have the same EC number and enzymatic function. So, these mentioned assignments could still be true, even if in UNIPROT and BRENDA no experimental data for these low-identity proteins are mentioned. Due to the limited manpower in UNIPROT and BRENDA only a small subset of papers can be annotated.

**Have the authors made all data and (if applicable) computational code underlying the findings in their manuscript fully available?**

Reviewer #1: Yes

Reviewer #2: Yes

Reviewer #3: Yes

Reviewer #4: Yes

PLOS authors have the option to publish the peer review history of their article (what does this mean?). If published, this will include your full peer review and any attached files.

Reviewer #1: No

Reviewer #2: No

Reviewer #3: No

Reviewer #4: **Yes: **Dietmar Schomburg
---

## [Decision Letter · Decision Letter 1]

13 Sep 2021

Dear Dr. Engqvist,

We are pleased to inform you that your manuscript 'Experimental and computational investigation of enzyme functional annotations uncovers misannotation in the EC 1.1.3.15 enzyme class' has been provisionally accepted for publication in PLOS Computational Biology.

Best regards,

Marco Punta

Associate Editor

PLOS Computational Biology

Arne Elofsson

Deputy Editor

PLOS Computational Biology

Reviewer's Responses to Questions

**Comments to the Authors:**

Reviewer #1: The authors have addressed the issues that I raised in my earlier review, and have made changes to the manuscript that I think are appropriate. I thank them for the careful attention they gave to the points I made.

Reviewer #2: The authors took into account essentially all my comments. No further questions

Reviewer #3: The authors have done a thorough job addressing the concerns raised in my review. It is important for the general reader that there is a better approach to presenting the expectations when examining enzymes with the same E.C. number.

**Have the authors made all data and (if applicable) computational code underlying the findings in their manuscript fully available?**

Reviewer #1: Yes

Reviewer #2: None

Reviewer #3: Yes

PLOS authors have the option to publish the peer review history of their article (what does this mean?). If published, this will include your full peer review and any attached files.

Reviewer #1: No

Reviewer #2: No

Reviewer #3: No

---

## [Editor Report · Acceptance letter]

17 Sep 2021

PCOMPBIOL-D-21-00508R1 

Experimental and computational investigation of enzyme functional annotations uncovers misannotation in the EC 1.1.3.15 enzyme class

Dear Dr Engqvist,

I am pleased to inform you that your manuscript has been formally accepted for publication in PLOS Computational Biology. Your manuscript is now with our production department and you will be notified of the publication date in due course.

With kind regards,

Amy Kiss
